# Exact Visualization of Deep Neural Network Geometry and Decision Boundary

**Ahmed Imtiaz Humayun**                                        IMTIAZ@RICE.EDU
*Rice University*
**Randall Balestriero**                                        RBALESTRIERO@META.COM
*Meta AI Research, FAIR*
**Richard Baraniuk**                                        RICHB@RICE.EDU
*Rice University*

**Editors:** Sophia Sanborn, Christian Shewmake, Simone Azeglio, Arianna Di Bernardo, Nina Miolane

## Abstract

Visualizing Deep Network (DN) geometry and decision boundaries remains a key challenge even today. In fact, despite the dire need for such methods e.g. to assess the quality of a trained model, to compare models, to interpret decisions, the community at large still relies on crude approximations. For example, computing the decision boundary of a model, say on a 2d slice of their input space, is done through gradient descent and sampling with dichotomy search. In this paper, we lean on the rich theory of Continuous Piece-Wise Linear (CPWL) DNs to provide, for the first time, a method that provably produces the exact geometry (CPWL partition) and decision boundary of any DN employing nonlinearities such as ReLU, Leaky-ReLU, and max-pooling. Using the proposed method, we are able to not only visualize the decision boundary but also obtain its spanning space, i.e., we can sample arbitrarily many inputs that provably lie on the model's decision boundary, up to numerical precision. We explore how such methods can be used to interpret architectural choices e.g. using convolutional architectures instead of fully-connected neural networks.

## 1. Introduction

Deep learning and in particular Deep Networks (DNs) have redefined the landscape of machine learning and pattern recognition. Although current DNs employ a variety of techniques that improve their performances, their core operation remains unchanged, mostly consisting of sequentially mapping an input vector $\boldsymbol{x}$ to a sequence of $L$ *feature maps* $\boldsymbol{z}^\ell$, $\ell = 1, \ldots, L$ by applying successive simple nonlinear transformations, often coined *layers*, as in

$$\boldsymbol{z}^{\ell+1} = \boldsymbol{a}\left(\boldsymbol{W}^\ell \boldsymbol{z}^\ell + \boldsymbol{b}^\ell\right), \quad \ell = 0, \ldots, L-1 \tag{1}$$

with $\boldsymbol{z}^0 = \boldsymbol{x}$, $\boldsymbol{W}^\ell$ the weight matrix, $\boldsymbol{b}^\ell$ the bias vector, and $\boldsymbol{a}$ an activation operator that applies a scalar nonlinear activation function $a$ to each element of its vector input. The parametrization of $\boldsymbol{W}^\ell, \boldsymbol{b}^\ell$ controls the type of layer employed e.g. circulant matrix for convolutional layer.

Interpreting the geometry of a DN is not a trivial task since many different parameters can lead to the same input-output mapping. A common example is obtained by permuting the

rows of $\boldsymbol{W}^\ell, \boldsymbol{b}^\ell$ and the columns of $\boldsymbol{W}^{\ell+1}$ any number of layer(s). It is clear that while overall mapping remains unchanged the parameters of the network differ. As a result, practitioners have relied on different solutions to interpret what has been learned by a model by looking at activations instead of looking at the weights of the network (Yosinski et al., 2015; Jalwana et al., 2021). One important method for model interpretation is to find the closest point to a training sample $\boldsymbol{x}$ that lies on the model's decision boundary (Somepalli et al., 2022), which finds practical use for active learning (Locatelli et al., 2018) and adversarial robustness (He et al., 2018). In this setting, one commonly performs gradient updates from an initial guess $\boldsymbol{x}$ based on an objective reaches its minimum whenever its argument lies on the model's decision boundary. Although alternative and more efficient solutions have been developed, most of the progress has focused on providing more optimized losses and sampling strategies (Somepalli et al., 2022; He et al., 2018). In short, there still exists a strong need to develop an exact method, up to machine precision, that is able to compute the decision boundary of a given DN.

In this paper, we focus on the family of DNs obtained by restricting $\boldsymbol{a}$, the activation functions, to be Continuous Piece-Wise Linear (CPWL) as is the case with the eponymous (leaky-)ReLU, absolute value, max-pooling. In this setting, the entire DN will itself become a CPWL operator, and we will demonstrate how from this observation alone, it is possible to probe the DN's geometry, and in particular the DN's decision boundary, in an exact manner. Echoing our previous example, such precise characterization will enable use to, e.g., sample arbitrarily many samples that lie on the DN's decision boundary, opening new avenues for interpretability. Furthermore, leaning on the CPWL form of the DN, we demonstrate that its geometry can be entirely described by its *input space partition* and *per-region affine mappings* which again, we will be able to obtain in closed-form. All in all, we provide a tractable and efficient method to visualize and quantitatively interpret the geometry of Deep Neural Networks, opening new doors to interpretability and visualization.

## 2. The Geometry and Decision Boundary of Continuous Piece-Wise Linear Deep Networks

The goal of this section is to first introduce basic notations and concepts associated with CPWL DNs (Sec. 2.1), and then develop our method that consists of building the exact DN input space partition, and the DN's decision boundary that lives on it (Sec. 2.2); empirical studies based on our method will be provided in Sec. 2.3.

### 2.1. Deep Networks as Continuous Piece-Wise Linear Operators

One of the most fundamental functional form for a nonlinear function emerges from polynomials, and in particular, spline operators. In all generality a spline is a mapping which has locally degree $P$ polynomials on each region $\omega$ of its input space partition $\Omega$, with the additional constraints that the first $P-1$ derivatives of those polynomials are continuous throughout the domain i.e. imposing a smoothness constraint when moving from one region to any of its neighbor.

More formally and for the context of DNs we will particularly focus on affine splines, i.e., spline operators with $P = 1$ and only constrained to enforce continuity throughout the

domain. In this setting, the partition $\Omega$ of the DN's domain $\mathbb{R}^S$, along with per-region affine mapping parameters $(\boldsymbol{A}_\omega, \boldsymbol{b}_\omega)$ corresponding to the *slope* matrix and *offset* vector respectively, enable to express the entire input-output mapping of a DN $\boldsymbol{S}$ as

$$\boldsymbol{S}(\boldsymbol{x}) = \sum_{\omega \in \Omega} (\boldsymbol{A}_\omega \boldsymbol{x} + \boldsymbol{b}_\omega) \mathbb{1}_{\{\boldsymbol{x} \in \omega\}}, \tag{2}$$

with $\boldsymbol{A}_\omega, \boldsymbol{b}_\omega$ being different for each region $\omega \in \Omega$ yet fulfilling the continuity constraint on $\boldsymbol{S}$, i.e., $\boldsymbol{S} \in \mathcal{C}^0(\mathbb{R}^S)$. The charming property of DNs is that those mappings and the partition are defined implicitly through the composition of affine and nonlinear operators and thus continuity always holds and unconstrained optimization of the DNs' parameters can be employed. Such formulations of DNs have been extensively employed as it makes theoretical studies amenable to actual DNs without any simplification while leveraging the rich literature on spline theory, e.g., in approximation theory (Cheney and Light, 2009), optimal control (Egerstedt and Martin, 2009), statistics (Fantuzzi et al., 2002) and related fields.

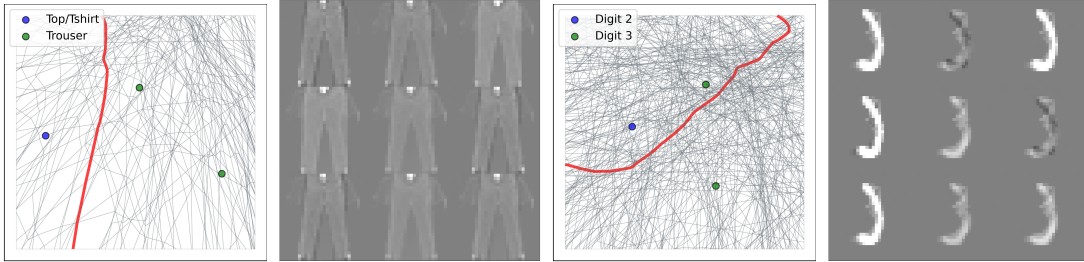

Figure 1: **(Left)** Decision boundary visualization for an MLP with width 50 and depth 3, trained on fashion-MNIST. Dark red line represents the learned decision boundary while black lines represent the spline partition of the network. **(Middle Left)** Samples from the decision boundary between classes Top and Trouser. The samples have distinguishable attributes present from both classes. **(Middle Right)** Decision boundary and partition visualization of a convolutional neural network trained on MNIST, with two convolutional layers and one hidden fully connected layer of width 50. One of the digit 3 samples is misclassified by the network as digit 2. **(Right)** Samples from the decision boundary between digits 2 and 3 of MNIST. Wall time required for finding the partitions is 8 min 16 s for the MLP and 21 min 9 s for the CNN.

## 2.2. Exact Computation of Their Partition and Decision Boundary

The key property that we will be leveraging in this section to obtain the decision boundary of a DN, is that given its partition $\Omega$ and the per-region affine mappings, adding an extra linear classifier on top exactly amounts as having a linear classifier within each region $\omega \in \Omega$ of the DN's input space. To see that, notice that the linear classifier is linear w.r.t. the output of $\boldsymbol{S}$, and that within each region of its partition, $\boldsymbol{S}$ is also linear. Although the decision boundary is nonlinear when looking at the whole domain, it becomes linear when restricted to different regions $\omega$ of that domain.

As a result, to visualize the decision boundary of a DN, we must first find the spline partition induced by its architecture and parameters, to then project the decision boundary to the input space within each of those regions. Based on this finding, we propose to first develop an optimized algorithm that will provide one with the partition $\Omega$ of a given DN which applies even if employing convolution and/or residual layers (Balestriero and Baraniuk, 2020). Let the DN be represented as a composition of $L$ affine layers, with per-layer affine parameters $\{\boldsymbol{W}^\ell, \boldsymbol{b}^\ell\}_{\ell=0}^{L-1}$. For ease of discussion, suppose the network has a ReLU non-linearity which we denote as $\sigma$ and thus each layer produces its output given some input $\boldsymbol{z}^{\ell-1}$ via

$$\boldsymbol{z}^\ell = \sigma(\boldsymbol{W}^\ell \boldsymbol{z}^{\ell-1} + \boldsymbol{b}^\ell).$$

Note that our implementation is easily extended to activation functions that are arbitrary affine splines e.g. ReLU, Leaky-ReLU, sawtooth. Each layer $\ell$ mapping can be considered a projection of incoming vectors $z^{\ell-1}$ onto a set of hyperplanes in the input space of the layer, defined by $\langle \boldsymbol{w}_i^\ell, z^{\ell-1} \rangle + b_i^\ell = 0$, where $\boldsymbol{w}_i^\ell$ is the i-th element of $\boldsymbol{W}^\ell$ and $b_i^\ell$ is the i-th element of $\boldsymbol{b}^\ell$. Each hyperplane projection gives the pre-activation of that layer for the corresponding output dimension. For a subsequent ReLU activation, vectors from the negative half-space created by the hyperplane are mapped to zero, while vectors from the positive half-space are mapped linearly. The key observation that one should notice is that the activations are linear except at 0. Therefore, for each layer, we have a set of hyperplanes in the input space of the layer which define the position of the non-linearities. Building on this intuition, for each layer our algorithm finds the set of convex polytopes formed by the intersection of hyperplanes, where each polytope represents a linear region. The pseudocode for finding the complete spline partition and the decision boundary can therefore be summarized as below:

- Given a bounded input domain, iterate through the first layer ($\ell = 1$) hyperplanes $\langle \boldsymbol{w}_i^\ell, \boldsymbol{z}^\ell \rangle + b_i^\ell = 0$ and cut the input domain to form convex polytopes. For each polytope $\omega$ the operation performed by the layer is an affine operation via parameters $\boldsymbol{A}_\omega^\ell = \boldsymbol{q}_\omega^\ell \odot \boldsymbol{W}^\ell$ and $\boldsymbol{b}_\omega^\ell = \boldsymbol{q}_\omega^\ell \odot \boldsymbol{b}^\ell$, where $\boldsymbol{q}_\omega^\ell$ is the activation pattern corresponding to the $\omega$ polytope of the $\ell$-th layer. Project the polytopes to the next layer via region-wise affine operation $\boldsymbol{z}^{\ell+1} = \boldsymbol{A}_\omega^\ell \boldsymbol{z}^\ell + \boldsymbol{b}^\ell$

- For layers $\{1, 2...L-2\}$ use the layer hyperplanes to partition the incoming polytopes and update the region-wise affine parameters. For each new region $\omega$ in layer $\ell$ input formed by cutting region $\omega'$ from layer $\{\ell' = \ell - 1\}$ output, the affine parameters will be $\{\boldsymbol{A}_\omega^\ell, \boldsymbol{b}_\omega^\ell\} = \{\boldsymbol{q}_\omega^\ell \odot \boldsymbol{W}^\ell \boldsymbol{A}_{\omega'}^{\ell'}, \boldsymbol{q}_\omega^\ell \odot \boldsymbol{b}^\ell + \boldsymbol{q}_\omega^\ell \odot \boldsymbol{W}^\ell \boldsymbol{b}_{\omega'}^{\ell'}\}$.

- For networks trained with softmax output activations, during inference we can consider the output activation as a max since the decision boundary between classes occurs when the $\arg\max$ differs. Therefore, the decision boundary between classes $\{i, j\}$ at the final layer input can be expressed by the hyperplane $\langle \boldsymbol{w}_i^\ell - \boldsymbol{w}_j^\ell, \boldsymbol{z}^\ell \rangle + b_i^\ell - b_j^\ell$. We therefore project the null space of the hyperplanes in layer $L-1$ to the input space to get the decision boundaries in the input space.

While the above method is generalized for arbitrary input dimensionality, it can be computationally expensive. In particular, it is well known that the number of regions with a

| Architecture | Dataset | Parameters | Avg Vol | Avg Number of Vertices | Ecc | Number of Regions |
|---|---|---|---|---|---|---|
| MLP | MNIST | 44,860 | 3.144e-4 | 3 | 102e7 | 318 |
| | Fashion-MNIST | 44,860 | 4.991e-4 | 3 | 36e7 | 1364 |
| CONV | MNIST | 39,780 | 1.134e-5 | 3 | 17e7 | 8814 |
| | Fashion-MNIST | 39,780 | 3.54e-5 | 3 | 14e7 | 28222 |

Table 1: Statistics of the spline partitions formed by fully-connected (MLP) and Convolutional neural networks. For each dataset, the same 2D slice and input domain is used to find the partition regions. Convolutional neural networks form a significantly higher number of regions compared to MLPs even with less parameters. The mean eccentricity and volume across regions is also significantly lower for convolutional neural networks.

partition $\Omega$ and the complexity of each region $\omega$ e.g. its number of faces and vertices, grow exponentially with respect to input dimensionality, even for the simplest one layer DNs. Therefore, in our implementation, given three points $x_1, x_2$ and $x_3$ we can define 2D slice of the input space through those points. We consider a bounded domain on this slice to compute the partition boundary. For the examples presented in Fig. 1, we consider two different classes for each network. We pick two of the closest training samples from the classes, along with a third point that is the nearest neighbor to one of the samples chosen. Therefore, we consider a square domain centered on the centroid of the three points.

### 2.3. Impact of Architecture on Partitions Properties

Computing the exact partition boundary finds many applications, not only to visualize and sample the decision boundary (see Fig. 1). We explore some alternative interesting directions in this section.

First, we explore the impact of the DN's architecture. We see that the choice of architecture can have significant effect on the partitioning induced by a deep neural network (Tab. 1). For a given dataset, we fix the input domain and switch between convolutional and fully connected architectures to draw emphasis on the effect of the symmetries induced by a convolutional layer. We see that in convolutional architectures, there is a significantly higher number of partition regions formed, which is an indication of higher complexity of the learned model (Montufar et al., 2014). We also see that the eccentricity and volume of the polytopes are significantly smaller for convolutional architectures compared to fully connected architectures, indicating more uniform partition shapes and higher partition density. These can also be visualized in Fig. 1.

### 3. Conclusions

We present the first provable method to visualize and sample the decision boundary of deep neural networks with CPWL non-linearities. Our presented methods may allow many future avenues of exploration and understanding of neural network geometries.

## Acknowledgements

Humayun and Baraniuk were supported by NSF grants CCF1911094, IIS-1838177, and IIS-1730574; ONR grants N00014-18-12571, N00014-20-1-2534, and MURI N00014-20-1-2787; AFOSR grant FA9550-22-1-0060; and a Vannevar Bush Faculty Fellowship, ONR grant N00014-18-1-2047.

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
