# OpenReview forum: "Exact Visualization of Deep Neural Network Geometry and Decision Boundary"
_NeurIPS.cc/2022/Workshop/NeurReps — NeurReps 2022 Poster_

### Official Review · Reviewer_wMUA · 2022-10-14
**An interesting visualization tool without any clear applications**

**Confidence:** 3
**Soundness:** 3
**Presentation:** 2
**Contribution:** 2
**Overall Rating:** 4

**Summary:**

The paper presents a computational method to visualize (in input space), the decision boundaries of a network which is performing classification. The method works for any deep feedforward network which has a continuous piecewise linear activation function, eg ReLU. The authors then apply the method to an MLP and a CNN trained to classify MNIST and fashion MNIST and report their findings.

**Questions:**

I have several questions for the authors. These are simple things the author can comment on or perform simple experiments for and this would have greatly increased the value of their work and my score. Perhaps the authors can do this in the future.

What are potential applications of this method? How can it be made useful to the community, beyond just as a visualization tool? Here I list out a few things that came to mind as I was writing my review which I hope the authors find constructive:

- Could be useful to look for adversarial examples, for instance?

- Could it help in evaluating other failure modes of classifier networks? Eg: representational collapse in self supervised learning.

- Can this visualization process be done over the course of training to understand something about the learning dynamics of a classifier?

- Can this method be used to understand why certain tricks make better classifiers? Eg: batchnorm, layernorm

- In the self-supervised learning setting, can this method shed light on the difference in representations obtained via say a contrastive loss vs a non-constrastive loss?

**Limitations:**

I have listed several suggestions for the authors to use their method for. As the paper stands, it is hard to evaluate the potential value of the authors' method.

**Recommended Decision:**

2: Borderline

**Relevance:**

2: Limited relevance

**Strengths And Weaknesses:**

Strengths:
The method is easy to understand, exact and works for any feedforward network with a CPL activation function.

Weaknesses:

- The figure is hard to understand. There are no axes labels and I don't understand what the grey lines are.

- In the table, it is commented that CNNs find a higher number of smaller regions. This is attributed to the symmetries present in the architecture. I don't understand intuitively why this is surprising or not or why this is a useful property. I would imagine the depths of the CNNs and MLPs are important but there is no information about this or comparisons across a variety of architectures with varying depth or width. The authors have only stated their findings without any conclusions. I find it very hard to evaluate the usefulness of their method.

**Submission Track:**

Extended Abstract (4 Page)

---

### Official Review · Reviewer_KGpU · 2022-10-14
**Interesting idea, could help with intuition for ReLU theory, practical impact unclear**

**Confidence:** 4
**Soundness:** 3
**Presentation:** 3
**Contribution:** 2
**Overall Rating:** 6

**Summary:**

This submission proposes to compute exactly the decision boundaries of neural networks with continuous piecewise linear activations on two-dimensional input slices. The technique used allows not only the decision boundary to be computed, but a complete mesh such that the network has affine response on each face. Statistics of this mesh, together with some visualizations on MNIST data are presented.

**Questions:**

01. What do you mean by the “geometry” of a deep network (in the title “visualization of DNN geometry”, or on page 2 “probe the DN’s geometry in an exact manner”) ? The use of the term “geometry” here is so broad it appears meaningless. The construction of the spline mesh and its visualization is arguably a geometric point of view, but it’s not so much the geometry of the network (by which some authors mean the connection geometry, or structure of the computational graph for instance) as it is the geometric structure of the learned function.

02. Are there any relevant insights to be drawn from the decision boundary in two dimensions on high-dimensional classification scenarios ? As is apparent on the illustrations of figure 1, samples on the decision boundary are essentially meaningless sums of images. This is typically comparable with optimal transport, where the mixture interpolation between two distributions is not particularly meaningful, and the intuitive “in between” sample corresponds more to a Wasserstein interpolation minimizing the transport distance rather than the $\ell_2$ norm. We could similarly imagine different more meaningful interpolations between samples, but wouldn’t that break this method, which requires critically that the input is a vector space (to use lines) of low dimension (to curb cost) ? How would you expect this technique to be used ?

03.  Why did you choose to plot only samples on the boundary in figure 1 ? It seems to me that, on the contrary, it would be more interesting to plot samples along the trajectory from one of the reference points to another, with the sample on the boundary in between, to observe the permanence or change of the visual features between samples on the trajectory. But then again, wouldn’t we just see a transparency-sum of the two endpoints, with some arbitrary meaningless image in the middle for the boundary, and is the boundary visually relevant ?

**Limitations:**

The authors do not directly address the limitations of this work, which is not considered a problem due to the very short format.

**Recommended Decision:**

3: Accept

**Relevance:**

3: Solid fit

**Strengths And Weaknesses:**

The ideas presented are clear, put simply, intuitive, and the applications chosen to illustrate the technique presented are easy to understand. The informal definitions in the exposition are a little too weak for a self-contained paper, and rely on the prior knowledge of the reader about usual notations instead of giving proper references that would introduce these notations, although this would be a very minor flaw for a paper that is not particularly theory-leaning. Being an extended abstract, it would seem odd to count the following as a weakness, but had this been a paper, it would be critically lacking an analysis of computational cost (and comparison with approximate methods) and at least one illustration of valuable insights that can be drawn from these visualizations to justify their study at all. Since it is a work in progress, I see no reason to object to the publication of this submission, but I will list all my concerns as though it were a full paper, in hope that this will help authors preemptively address similar questions future reviewers and readers might have, when this work is eventually extended into a longer publication.

#### **Detailed comments**

01.   The choice of axes on figure 1 seems questionable. I could not find a description of this choice in the text, only that three points are chosen to define a plane, which would allow any affine transformation a posteriori, why not at least use a shear to make full use of the square in the illustration ? Preservation of volumes does not seem critical since they’re not easily interpretable anyway, and the choice of the picture is inconsistent with the definition for the axes of $S$ in page 4.

02. The discussion of computational cost is entirely missing. The choice of exact vs approximate methods usually boils down to the balance between usefulness of the precision versus cost to obtain the given precision. Given that I have no argument in favor of particularly precise boundaries, I would expect to never use exact methods unless they are comparatively cheaper than approximate methods, which could largely explain why this technique as not been introduced before.

03. Related to the previous concern, the overhead introduced by the computation of the spline mesh, as opposed to only the decision boundary, is not addressed anywhere. I can see why computing the mesh and its evolution over training could be helpful to better understand the functions learned by ReLU-based networks, but this submission seems centered on the decision boundary (starting with its title), and I would intuitively guess that computing the mesh is orders of magnitude more expensive that only the decision boundary.

04. Any conclusion on the basis of table 1 alone seems abusive. At the very least, to draw conclusions as broad as “convolutional networks form a significantly higher number of regions compared to MLPs”, one would expect several repetitions over random initializations showing consistency of the statistics observed, and comparisons across various choices of hyperparameters for each architecture since those are chosen arbitrarily for the experiment.

05. In the introduction, “the core definition [of DNs] has changed much” is not supported by references or examples of other definitions. For instance the definition of the cognitron by Kunihiko Fukushima in 1975 is almost identical, except for a division before the non-linearity, which would break the here-crucial property of piecewise linearity.

06. In the introduction, no types are given for the input, weights, biases, or feature maps, and no signature for the activation. Adding just $z_l \in \mathbb{R}^{n_l}$ would clarify the notation for readers with backgrounds not focused on deep learning, and make apparent the choice of a width for each layer (see problems with Table 1 hyperparameters hereafter).

07. There seems to be a typo on the bottom of page 2: “Let, our network can be represented as”.

08. There is no definition of V-polytope and no reference. A short explanation could also be a good place to discuss the representation in memory as vertices rather than hyperplanes, for instance.

09. On page 2, “alternative methods have been developed” gives no references and no names for such methods.

10. On the top of page 2, there is a typo in “the entire DN will itself becomes”, there is no ‘s’ for the future “will become”.

11. On the bottom of page 2, “residual blocks” are not affine operations. Convolutions are affine, but residual blocks (with CPWL non-linearities) are only piecewise affine.

12. In Table 1, why is the average number of vertices an integer every time ? If it is constant, that is a more interesting information than the average number alone, and if not it should be represented with decimals in the table.

13. In Table 1, the number of significant digits displayed varies across rows for the volume. The eccentricity column has the same problem, and misplaces the decimal in the scientific notation.

14. In the caption for Table 1, the claim that eccentricity is significantly lower in convolutional networks seems misplaced. Is it really “significantly” lower, how is significance appreciated here ? Is this claim really supported by a single experiment without repetitions or variations across even network widths ? Is this change in eccentricity relevant or intepretable in any way ? How is the eccentricity defined and why is it greater than one ? Discussions of this measure and its significance would be welcome, and more appropriately located in the main text.

15. On the bottom of page 4, “the choice of architecture can have significant effect on the partitioning induced by a deep neural network” is either trivial (changing widths affects partitioning) or too broad for the evidence supporting it (convolution vs full-connection has only one repetition, one choice of depth, and the widths are not even specified).

16. What does “first provable method” refer to in the conclusion, what is being proved ? Is it used to mean “provably correct method”, in the sense that the result returned is provably close to the real decision boundary, and if so why is this the first, since sampling-based methods are also provably correct in the asymptotic regime.

**Submission Track:**

Extended Abstract (4 Page)

---

### Official Review · Reviewer_gZzd · 2022-10-16
**review of Exact Visualization of Deep Neural Network Geometry and Decision Boundary**

**Confidence:** 3
**Soundness:** 3
**Presentation:** 3
**Contribution:** 3
**Overall Rating:** 7

**Summary:**

This extended summary proposes to propose an explore the partition and decision boundaries of deep networks using piecewise linear activation functions, for visualization purpose. The proposed approach is to find the convex polytopes partitioning the input space, seen as a set of hyperplanes.


**Questions:**

It could be interesting to make a visual comparison with methods that estimate decision boundaries through sampling, to illustrate the difference between estimated and exact methods.


**Limitations:**

The computational load (or time) could be mentioned in the Table 1 to provide an estimation of the direct applicability of the approach.


**Recommended Decision:**

3: Accept

**Relevance:**

4: Highly relevant

**Strengths And Weaknesses:**

This works seems original with a novel approach to determine decision boundaries with an exact approach, whereas other approaches are mostly sampling methods. This seems a sound theoretical idea, that is supported by some examples with small deep net and conv net. It extends the work done by Balestriero & Baraniuk.

The main limitation is the time complexity, as it seems that the method becomes quickly computationally expensive for high-dimensional input. The authors restrict their analysis to triplets of samples to compute the partition boundary.


**Submission Track:**

Extended Abstract (4 Page)

---

### Decision · Program_Chairs · 2022-10-21

Accept (Poster)